# Correlation between Vegetation Structure and Species Diversity in Traditional Villages in Karst Topographic Regions of the Zunyi City, China

**DOI:** 10.3390/plants11223161

**Published:** 2022-11-18

**Authors:** Caijie Deng, Zongsheng Huang, Xiaojing Zhang, Hua Zhao, Siyu Jiang, Yuxin Ren

**Affiliations:** 1College of Forestry, Guizhou University, Guiyang 550025, China; 2College of Architecture and Urban Planning, Guizhou University, Guiyang 550025, China; 3College of Construction Engineering, The College of Humanities and Science of Guiyang, Guiyang 550025, China

**Keywords:** plant community structure, species diversity, community structure-species diversity relationship, traditional villages in karst regions, Zunyi City

## Abstract

Studying the relationship between vegetation structure and diversity is important in an area having karst topography and unique traditional customs. We selected a total of six traditional villages in Zunyi City, China, to collect vegetation data. Additionally, using one-way ANOVA and the Pearson correlation coefficient analytic method to analyze, the results showed that, overall, plant communities were mostly regularly distributed. The overall differentiation degree was low aggregation, intensity, and the extreme intensity mixed state. Overall, competitive pressure, growth vigor and stability were better than the natural forest. The community stability index at lower altitude was significantly higher than that at higher altitude. The recorded plant communities in the living space were typically aggregated, the plant communities were randomly distributed in the production space, and the plant communities were uniformly distributed in the ecological space. In general, the diversity indexes, except the *Jh* index, were the highest in the herb layer; the second was in the shrub layer and the lowest was in the tree layer. Species diversity at the middle altitude was higher than that at low and high altitudes (except for the shrub at a high altitude of 1100–1160 m). The overall plant species diversity was highest in the living space, second highest in the ecological space and lowest in the production space. On the whole, there was a significant correlation between the spatial structure of plant communities and the species diversity of plant communities at different altitudes, and in *PLE* spaces. The main objective of this study was to reveal the plant community structure, species diversity, and their relationship under the dual effects of national traditional culture and karst landform. Additionally, we sought to provide theoretical guidance for the construction of plant community protection and biodiversity conservation in traditional villages in karst areas.

## 1. Introduction

Traditional villages, also termed ancient villages, with both material and intangible cultural heritage [1], have important historical, cultural, scientific, artistic, social, and economic value [2]. However, with the rapid advancement of industrialization and urbanization, the homogenization of rural plant landscapes is becoming more and more serious [3]. The plant landscape is the soul of traditional villages, as well as an important embodiment of their landscape features and cultural connotations. Plant community structure and species diversity are the basis for representing the function and process of the rural plant landscape. Community structure determines the niche and competition pattern of species in the community and the trend of community succession [4]. At this stage, the tracking of stand spatial structure has always been an important aspect of natural forest management simulation technology in various countries [5]. Since Zhao et al. [6] proposed to use the target tree and its surrounding four neighboring trees as the basic object of study, a large number of studies have been conducted to analyze the spatial structure characteristics of different stand types and regions, and to apply them to the spatial measurement of species diversity, the competitive pressure of trees, and the restoration and reconstruction of stand structure [7]. Species diversity is a comprehensive reflection of species richness and the uniformity of distribution [8]. Measuring the complexity and stability of community functions [9] has long been a hot topic of research in community ecology [10]. Meanwhile, due to the influence of urbanization, traditional farming methods and woodland structure have changed, destroying the plant species diversity of traditional villages. Therefore, it is important to study the spatial structure characteristics of traditional village plant communities, and the relationship between species diversity. To reveal the formation mechanism of village plant community structure, the conservation and development of village plant landscapes and the construction of an ecologically pleasant living environment should be coordinated, to ensure social, economic, and ecological benefits under the revitalization of the countryside.

Early research focused on the spatial characteristics and cultural value of traditional villages [11,12]. Community structure focuses on the spatial distribution pattern of species [13], the construction of the parameter system of stand spatial structure [14], and the effect of stand spatial structure on maintaining the growth and community stability of trees in a stand [15]. Species diversity mainly focuses on the measurement of grassland plant diversity [16] and the species diversity of a broad-leaved forest [17]. However, there are few studies on rural plant community structure and species diversity. Studies on the relationship between community structure and species diversity are mainly focused on large scale areas, such as natural forests and economic forests areas [18]. There are few reports on the study of small-scale and natural rural plant communities, especially on the biodiversity and stability of traditional villages. With the introduction of the national rural revitalization strategy and other policies, the construction of the rural ecological environment has become a top priority, but the typical plant landscape characteristics of its ecological environment have hardly been studied systematically. Currently, village plant landscape construction is based on urban plant landscape theory; the rural landscape is excessively urbanized.

Based on this, this study takes the traditional village in the karst topographic regions of Zunyi City as the object of study. A total of 39 traditional villages in Zunyi City are listed in the National Traditional Villages List, which is also a typical karst topography. Zunyi City is a key area for economic development and ecological construction in the Yangtze River economic belt of China, which is of great value for humanistic, economic and ecological research. A study showed that there are 170 families of seed plants in Zunyi City, containing 982 genera and 3596 species, with rich plant species [19]. The aim of this study is to reveal the structure, species diversity, and the relationship between the traditional culture of the people and the karst landscape in order to expand the direction of flora research, to deepen the theory of flora research, and to provide a theoretical basis for the maintenance of the native flora of traditional villages in Zunyi and the construction of rural flora in other karst areas. The aim is to provide a theoretical basis for the maintenance of traditional village botanical landscapes in Zunyi and the establishment of rural botanical landscapes in other karst areas, and thus provide theoretical guidance for the sustainable development of village biodiversity under the strategy of rural revitalization and urbanization in China.

## 2. Materials and Methods

### 2.1. Study Area

Zunyi is located in the north of Guizhou Province. The geographical location is 106°17′22″ E~107°26′25″ E, 27°13′15″ N~28°04′09″ N (Figure 1). With a total area of 30,780.73 km^2^, it belongs to a typical subtropical humid monsoon climate zone, with cold and wet winters and hot and rainy summers. At an altitude of 221–2227 m above sea level, the landform type is more complex, with obvious topographic undulations, gradually becoming gentler from west to east, with mountains, hills, basins, and river valley dams accounting for 65.08%, 28.35%, and 6.57% of the total area, respectively. The annual mean temperature is 15.1 °C and the annual mean precipitation is 998.6 mm. Soil types such as purple soil, limestone soil, rice soil, and loam, are mainly distributed in the study area, and the parent materials are mainly sandstone, shale, sand shale, limestone, and quaternary red clay. The city’s forest cover is 62% and the city is rich in species diversity, with 3820 species of higher plants in 1110 genera of 289 families and 42 species of wild plants protected at the national level, including 9 species of plants protected at the national level and 33 species of plants protected at the second level (http://www.lyj.zunyi.gov.cn/ (accessed on 28 October 2022)) including silver fir (*Cathayaargyrophylla*), spinulose (*Alsophila spinulosa*), red bean fir (*Taxus chinensis*), and dove (*Davidia involucrataBaill*) protected plants, etc. There are special protected areas in Zunyi and five vegetation types: coniferous forest, broad-leaved forest, scrub, aquatic vegetation, and meadow.

### 2.2. Research Methodology

#### 2.2.1. Village Selection and Overview

We selected six traditional villages for the study based on the principles of completeness, representativeness, and topographic homogeneity (Table 1). With topographic homogeneity meaning that all the villages are located in karst areas, and completeness meaning that all the villages have the following: the surrounding mountains are intact; the vegetation is intact; the landscape environment remains pristine and characteristic; and the villages basically maintain their traditional patterns and have a more harmonious overall appearance (Table 1 for a representative and basic overview of the sample villages).

#### 2.2.2. Determination of Parameter Values for Each Analytical Perspective

This study presents a comparative analysis of the plant community structure and species diversity in three aspects: the whole village, different altitude gradients, and the “production-living-ecological” spaces. To reveal the characteristics of community structure and species diversity, we took the average of all community parameters, to analyze the whole village, according to the altitude range of plant community distribution to divide the altitude gradient. An altitude gradient is set every 100 m from 600 m above sea level, with a total of five gradients: 600–700 m, 700–800 m, 800–900 m, 1000–1100 m, and 1100–1160 m. The “production-living-ecological” spaces and type of green spaces are based on the specific contents of the three growth spaces of production, living and ecology [20]. The research objects are divided into 11 types of green spaces: vegetable fields, tea gardens, farmlands and fruit forests in the production space; roads, squares, rituals, courtyards and houses in the living space; and forests and rivers in the ecological space. (Table 2 and Table 3).

#### 2.2.3. Sample Plot Setting and Survey

Fifty-seven sample plots were set up to analyze the spatial structure and species diversity of the plant communities. To better characterize each community type, each plot was 50 m × 20 m (the length and width of the sample plot could be adjusted according to the actual situation). Ten tree quadrats of 10 m × 10 m were set in each sample plot and 10 shrub quadrats of 5 m × 5 m were set in each tree quadrat, with a randomly set 1 m × 1 m herb quadrat in the shrub sample [21] (Figure 2). Using the conventional plant community survey method [22], all trees of height (H ≥ 3 m) in the sample plots were checked; recording one by one their species name, number of plants, diameter at breast height, height, and crown width. We recorded the species name, the number of plants, ground diameter, and the height of the shrub layer (H < 3 m, including young seedlings of understory trees). We recorded the species name, the number of plants, and the height and cover of the herb layer. We recorded basic information such as latitude, longitude, elevation, measured slope, slope direction, and slope position. Using the Winkelmass software and R3.5.1 to calculate the uniform angle index (*W*), dominance (*U*), and mingling (*M*) of individual stands with a diameter at breast height (*DBH*) ≥ 5 cm, we analyzed the characteristics of four indicators: the competition index (*CI*); forest growth vitality (*DC*); tree stability (*DH*); and community species diversity.

#### 2.2.4. Determination of Plant Community Spatial Structure and Non-Spatial Structure Parameters

Stand spatial structure refers to the pattern of distribution of trees on the stand, and the way their attributes are arranged in space [23]. In this study, the stand spatial structure parameter was used to characterize the spatial structure of the community. Additionally, *n* = 4, that is, the structure index was calculated for the distribution of the four nearest neighboring trees around the core tree, with a standard angle α_0_ = 72°. To avoid the influence of edge effects on the spatial structure parameters, a 5 m buffer zone was set, and the stands within the buffer zone were only involved in the calculation as adjacent trees (Figure 3 and Table 4).

According to Gadow and Hui [24], when $\bar{W}$ > 0.517, cluster distribution, 0.475 < *W* < 0.517, random distribution, and $\bar{W}$ < 0.475, uniform distribution, the community distribution pattern was determined accordingly in this study. The *CI_i_* is a negative indicator, the larger *CI_i_* indicates that the smaller the forest, the greater the competitive pressure. In the non-spatial structure parameters, the *DC_i_* is a positive indicator, the greater the *DC_i_*, the higher the vitality of forest growth. *DH_i_* is a negative index, the smaller the *DH_i_*, the more stable the forest.

#### 2.2.5. Determination of Plant Community Species Diversity Index

The community species diversity measurement method [27] is specifically as follows.

The Margalef Index:(1)$$\;R=\frac{\left(S-1\right)}{\ln N}$$

The Shannon-Wiener Diversity Index:(2)$$\;H=\overset{s}{\underset{i=1}{\sum }}P_{i}\ln P_{i}$$

The Simpson Index:(3)$$D=1-\sum_{i=1}^{s}{\left(P_{i}\right)}^{2}$$

The Pielou Evenness Index:(4)$$Jh=\frac{H}{\ln S}$$

In the formula: *p_i_* = *n_i_*/*N*, *N* is the sum of individuals of all species, *n_i_* is the *i*th species. ln is the natural logarithm based on *e*, and *S* is the total number of species.

### 2.3. Data Processing

The data were statistical analyzed using the Excel 19.0 and SPSS 22.0 software. By Winkelmass, the forest spatial structure parameters were calculated, and the differences between the different data groups (α = 0.05) were compared by one-way analysis of variance (*one-way ANOVA*) and the least significant difference method (*LSD*), and the correlation between the community structure and the diversity of species in the plant community was tested by Pearson correlation analysis (Figure 4).

## 3. Results

### 3.1. Analysis of Plant Community Structure in Traditional Villages in Karst Topographic Regions of the Zunyi City

#### 3.1.1. Analysis of the Overall Plant Community Structure of Traditional Villages in Karst Topographic Regions of the Zunyi

Table 5 shows that: 33.33% of the trees were aggregated; 28.07% were randomly distributed; 38.60% were evenly distributed; 50.88% were in a moderate state; and 45.61% were in an inferior state. In all, 52.63% were strongly mixed, followed by 22.81% which were very strongly mixed. The dominance of growth was moderate and in a moderately strong mixed state, indicating that the differentiation degree of *DBH* size of a single tree in the community was more balanced, the aggregation growth of the same tree species was less, and the isolation degree of the tree species was higher. The mean value of the overall *CI* was 1.715, indicating that the overall stand was under great competitive pressure. The mean value of *DC* was 0.333, indicating that the overall tree growth vigor was general. The mean value of *DH* was 0.645, indicating that the overall stand was relatively stable.

#### 3.1.2. Analysis of Plant Community Structure at Different Altitude Gradients

Table 6 shows that, at the altitude of 600–700 m, distribution was mainly aggregated, the uniform distribution was in the middle, and the random distribution was the smallest. The altitude of 700–1100 m was dominated by uniform distribution, while the altitude of 1100–1160 m was dominated by aggregation distribution, indicating that the altitude of 700–1100 m was greatly influenced by human factors. The degree of differentiation of the size of each altitude range was dominated by moderation and disadvantage, of which 600–700 m was dominated by the disadvantage, and 700–1160 m is dominated by the moderation, indicating that the lower the altitude, the more the degree of forest differentiation tends to be inferior, and vice versa. Intensity mixing was dominant in each altitude interval, followed by extreme intensity mixing. The *CI* decreased first and then increased with the increase of altitude, and the *CI* at medium altitude was significantly higher than that at lower and higher altitudes. Among the non-spatial structure parameters, the *DC* was the highest at 800–900 m (0.351), followed by 600–700 m, 1000–1100 m, and 1100–1160 m, and the lowest at 700–800 m (0.293). The *DH* value was the highest at the lowest altitude of 600–700 m (0.706), and the *DH* value of the lower altitude community was significantly higher than that of the higher altitude, indicating that the higher the altitude, the more stable the community.

#### 3.1.3. Analysis of the Plant Community Structure under the “Production-Living-Ecological” Spaces

Table 7 shows that the *W* was mainly randomly distributed in the production space, with uniform distribution in the middle and aggregated distribution in the smallest. It was mainly aggregated in the living space, with uniform distribution in the middle and random distribution in the smallest. It was mainly uniformly distributed in the ecological space, with aggregated distribution in the middle and random distribution in the second, which indicated that the living space was mainly influenced by human activities, while the production and the ecological space were mainly influenced by nature. The degree of size differentiation of each space was moderate, it indicated that the size differentiation of trees in the living space is good. The intensity of mixed interbreeding dominates in all spaces, followed by very strong mixed interbreeding, reflecting that there were only one to two trees of different species around most of the trees in each space, indicating that the trees in each space were well mixed and the spatial segregation of tree species was high. The *CI* was highest in the ecological space, middle in the living space and lowest in the production space, indicating that the competition among trees in the ecological space was the most intense and the *DH* was relatively poor. The *DH* was highest in the living space, middle in the ecological space and lowest in the production space, and the *DC* was highest in the living space, second in the production space and lowest in the ecological space, indicating that the community growth vigor was poor due to greater human disturbance in the ecological space, and vice versa in the living space.

**Table 5 plants-11-03161-t005:** Structural and non-structural parameters of the overall plant community of traditional villages in karst topographic regions of the Zunyi.

Parameter Name	Uniform Angle Index (W)			Dominance (U)					Mingling (M)					Competition Index (CI)	Growth Vitality (DC)	Stability (DH)
	W > 0.517	0.475 < W < 0.517	W < 0.475	0	(0, 0.25]	(0.25, 0.5]	(0.5, 0.75]	(0.75, 1]	0	(0, 0.25]	(0.25, 0.5]	(0.5, 0.75]	(0.75, 1]	——	——	——
Specific meaning	Aggregate distribution	Random distribution	Evenly distributed	dominance	sub-dominant	mean state	disadvantages	Absolute	Zero Mixing	Weakly mixed	Moderately mixed	Strength mixing	Very strong	——	——	——
Percentage	33.33%	28.07%	38.60%	--	3.51%	50.88%	45.61%	--	5.26%	8.77%	10.53%	52.63%	22.81%	1.715 ± 0.515	0.333 ± 0.132	0.645 ± 0.209

**Table 6 plants-11-03161-t006:** Structural and non-structural parameters of plant communities at different altitude gradients.

Altitude (m)	Uniform Angle Index (W)			Dominance (U)					Mingling (M)					Competition Index (CI)	Growth Vitality (DC)	Stability (DH)
	W > 0.517	0.475 < W < 0.517	W < 0.475	0	(0, 0.25]	(0.25, 0.5]	(0.5, 0.75]	(0.75, 1]	0	(0, 0.25]	(0.25, 0.5]	(0.5, 0.75]	(0.75, 1]	——	——	——
Specific meaning	aggregated distribution	random distribution	uniform distribution	dominance	sub-dominant	mean state	disadvantages	absolute disadvantage	zero degree mixed	weak mixed	moderate mixed	intensity mixing	very strong mixed	——	——	——
600–700	60.00%	10.00%	30.00%	——	——	20.00%	80.00%	——	——	20.00%	10.00%	50.00%	20.00%	1.913 ± 0.303a	0.334 ± 0.135a	0.706 ± 0.265a
700–800	27.27%	18.18%	54.55%	——	9.09%	72.73%	18.18%	——	9.09%	——	——	63.64%	27.27%	1.721 ± 0.600ab	0.293 ± 0.091a	0.539 ± 0.162b
800–900	11.11%	38.89%	50.00%	——	——	50.00%	44.44%	5.56%	——	——	16.67%	55.56%	27.78%	1.605 ± 0.421b	0.351 ± 0.172a	0.676 ± 0.185a
1000–1100	25.00%	33.33%	41.67%	——	8.33%	58.33%	33.33%	——	16.67%	25.00%	16.67%	33.33%	8. 33%	1.569 ± 0.675b	0.344 ± 0.104a	0.632 ± 0.242ab
1100–1160	57.14%	28.57%	14.29%	——	——	57.14%	42.86%	——	——	——	14.29%	42.86%	42.86%	1.917 ± 0.471ab	0.333 ± 0.126a	0.659 ± 0.155ab

**Table 7 plants-11-03161-t007:** Structural and non-structural parameters of the plant community under the “production-living-ecological” spaces.

“PLE Spaces”	Uniform Angle Index (W)			Dominance (U)					Mingling (M)					Competition Index (CI)	Growth Vitality (DC)	Stability (DH)
	W > 0.517	0.475 < W < 0.517	W < 0.475	0	(0, 0.25]	(0.25, 0.5]	(0.5, 0.75]	(0.75, 1]	0	(0, 0.25]	(0.25, 0.5]	(0.5, 0.75]	(0.75, 1]	——	——	——
Specific meaning	Aggregate distribution	Random distribution	uniform distribution	dominance	sub-dominant	mean state	disadvantages	Absolute	Zero Mixing	Weak mixed	Moderately mixed	intensity mixing	Very strong	——	——	——
produce	22.22%	44.44%	33.33%	——	22.22%	44.44%	33.33%	——	11.11%	11.11%	——	66.67%	11.11%	1.382 ± 0.472a	0.297 ± 0.100ab	0.500 ± 0.130a
Living	37.93%	27.59%	34.48%	——	——	48.28%	51.72%	——	——	3.45%	10.34%	48.28%	37.93%	1.707 ± 0.425ab	0.375 ± 0.152a	0.682 ± 0.210b
Ecology	26.32%	15.79%	57.89%	——	——	57.89%	36.84%	5.26%	10.53%	15.79%	21.05%	42.11%	10.53%	1.884 ± 0.611b	0.285 ± 0.092b	0.657 ± 0.216ab

### 3.2. Analysis of Plant Species Diversity in Traditional Villages in Karst Topographic Regions of the Zunyi

#### 3.2.1. Overall Plant Community Species Diversity in Traditional Villages in Karst Topographic Regions of the Zunyi

Table 8 show that all the diversity indices were highest in the herb layer (except for the *Jh*), second highest in the shrub layer and lowest in the tree layer. This was mainly due to the fact that the herb species are the most numerous, while there are more plantations or pure forests in the villages, resulting in a more homogeneous tree layer with the lowest species diversity.

#### 3.2.2. Species Diversity of Plant Communities at Different Altitude Gradients

Table 9 shows that the diversity of species at medium altitudes was higher than at lower and higher altitudes (except for the diversity of the shrub layer at higher altitudes in the range 1100–1160 m), while the diversity of species at lower altitudes in the range 600–700 m was lower. The high level of anthropogenic activities (including farming, woodcutting, and herb grubbing) was the main reason for the low diversity of species at this range. The diversity indices of the shrub layer do not fluctuate much, while the diversity indices of the tree and grass layers show significant differences across the altitude range, indicating that shrubs are more suitable for karst topographic habitats.

#### 3.2.3. Species Diversity Analysis of Plant Communities under the “Production-Living-Ecological” Spaces

Table 10 shows that the overall plant species diversity was highest in the living space, followed by the ecological space and lowest in the production space. Except for the tree layer, the shrub layer in the Pielou index and the herb layer in the Shannon-wiener index, the species diversity of the tree layer, the shrub layer and the herb layer in productive space was significantly lower than that in the living and ecological space. The diversity of species in the shrub layer was highest in the living space, followed by the ecological space and lowest in the production space, and the indices of the shrub layer, except for *J*, differed significantly in the different three spaces. The indices of the herb layer, except for the Margalef index, differed significantly in the different three spaces, indicating that the species diversity of different spatial types in karst topographic habitats was different. 

### 3.3. Correlation between Plant Community Structure and Species Diversity in Traditional Villages in Karst Topographic Regions of the ZUNYI City

#### 3.3.1. Relationship between Plant Community Structure and Species Diversity in Traditional Villages in Karst Topographic Regions of the Zunyi City as a Whole

Figure 5 exhibits that, on the whole, the relationship between species diversity and community structure in the herb layer was the closest, followed by the tree layer, and the relationship between the community structure index and the diversity of the shrub layer was very weak and not significant. Among the community structural and non-structural parameters, the most influential ones on species diversity in the tree layer were the *DC* and the *CI*, where the *DC* was significantly positively correlated with species diversity in the tree layer, indicating that the greater the growth vigor of the tree species, the higher the species diversity in the tree layer. The *CI* was significantly negatively correlated with species diversity in the tree layer *Jh*. The *CI* was an important structural parameter affecting the diversity of the shrub layer, and the *CI* was significantly and positively correlated with the shrub layer *R*, indicating that the size of the competitiveness of the shrub layer stands has a strong influence on the shrub layer diversity index. The common key factors affecting species diversity in the herb layer were W, *DC*, and *DH*, with *W* showing a highly significant negative correlation with the herb layer diversity indexes *D* and *Jh*, and a significant positive correlation with tree layer Jh, indicating that the more the spatial distribution pattern of the community tends to be aggregated, the lower the herbaceous layer species diversity index, and the more evenly distributed the tree layer species. Additionally, the correlation between the *U* and the *M* and the diversity indexes was the weakest, or not significant.

#### 3.3.2. Relationship between Plant Community Structure and Species Diversity in Traditional Villages at Different Altitude Gradients

Figure 6 shows that, overall, the M was significantly or highly significantly positively correlated with the species diversity of the tree layer in all elevation steps, except for 700–800 m. In addition, the *M* was highly significantly positively correlated with the species diversity of the shrub layer in the highest elevation 1100–1160 m, and was significantly positively correlated with the species diversity of the herb layer *R*. This indicates that in high altitude areas, the greater the mingling degree of tree layer, the higher the species diversity of the shrub and herb layer. The *W* had some effects on herb layer species diversity at the lowest and highest altitudes. The *W* had significantly negative correlation with the herb layer species diversity *D* at the lowest altitude of 600–700 m, and had significantly positive correlation with the herb layer species diversity *R* at the highest altitude of 1000–1160 m. At the lower altitude of 700–800 m, except for the angle scale and mingling degree, the other factors had a greater impact on plant species diversity, and the *U* was significantly negatively correlated with the tree layer species diversity *Jh*. The *CI* was significantly negatively correlated with the herb layer species diversity. The *DC* was highly significantly positively correlated with the tree layer species diversity and negatively correlated with herb layer *Jh*. The *DH* was significantly negatively correlated with the herb layer *H* and *D*. The *D* was significantly negatively correlated with the herb layer *H* and *D*.

#### 3.3.3. Relationship between Plant Community Structure and Species Diversity under the “Production-Living-Ecological” Spaces of Traditional Villages

Figure 7 shows that, overall, the plant community structure is most closely related to species diversity in the production space. Except for the *U* and *CI*, other structural factors were significantly or extremely significantly correlated with the species diversity of trees, shrubs, and herbs layer. Among them, the *W* was significantly or extremely significantly positively correlated with the species diversity of the shrub layer. The *M* was significantly positively correlated with species diversity in the shrub layer and significantly negatively correlated with the herb layer. The *DC* was significantly positively correlated with species diversity in the tree layer and significantly negatively correlated with the shrub layer. The *DH* was significantly positively correlated with species diversity in the tree layer and highly significantly positively correlated with the herb layer. In the living space, the largest impact on the species diversity of the herb layer is the *W*, which was significantly negatively correlated. In the ecological space, the key factors affecting the species diversity of the herb layer are *W* and *DH*, and the *W* was significantly or extremely significantly negatively correlated with the species diversity of the herb layer, indicating that the size of the forest distribution pattern has a greater impact on the species diversity of the herb layer.

## 4. Discussion

### 4.1. Plant Community Structure and Species Diversity Characteristics of Traditional Villages in Karst Topographic Regions of the Zunyi

The spatial structure of a stand determines the spatial arrangement of trees within a population and its ability to occupy resources in the surrounding environment, which can have an impact on the growth, stability, and the biodiversity of the stand [28]. Some studies have shown that the more stable the structure of a community, the more the spatial pattern of trees tends to be randomly distributed or less aggregated. In this study, the *W* scale of the plant communities in the traditional villages in karst topographic regions of the Zunyi city ranged from 0.430 to 0.603, partly close to random distribution. It can be seen that the community in the study area belongs to a higher mingling degree and is in the transition stage of succession to a stable state. This is similar to the results of Jiacheng Zhang et al. [29] and Tingting Chen et al. [30] on the spatial structure of broad-leaved evergreen forests and natural forests in their natural state [31], which indicate that plant communities in rural habitats can still show a high degree of mixing and random distribution characteristics consistent with their natural state and natural forests. This is due to the cultural traditions of traditional villages, whose inhabitants have a reverence for nature, even in the harsh ecological environment of karst areas. The overall competitive pressure is high, and growth vigor and stability are better compared to natural forests. Under different altitude gradients, the *DH* values of communities at lower altitudes are significantly higher than those at higher altitudes, indicating that the higher the altitude of traditional villages in karst areas, the more stable the communities are [32]. Among the “production-living-ecological” spaces, the communities are typically clustered in the living space, randomly distributed in the production space, and mainly evenly distributed in the ecological space, indicating that the residents of traditional villages in karst topographic regions have always maintained the concept of respecting and revering nature in their lives and a sound social system (e.g., the “payment” system, i.e., the maintenance of ecology by means of internal contracts). Under the impetus of relevant national policies, it has played an active role in maintaining the stability of plant communities in traditional villages in karst of the Zunyi.

Studies have shown that the understory environment is complex, and the diversity characteristics of understory herbaceous species are different [33]. This study showed that in the whole plant community, the diversity indexes were the highest in the herb layer, except *Jh*, the second was in the shrub layer and the lowest in the tree layer, which is consistent with the results of Zhang Jianyu et al. [34]. On the one hand, the management and utilization of local biodiversity by ethnic minorities has also resulted in a variety of traditional cultural practices and traditional knowledge. The collection and use of local biological resources for food, medicine, architecture, and costume by villagers is a direct manifestation of the richness of biodiversity. On the other hand, the traditional culture of ethnic minorities also contributes to the continuous renewal and preservation of biological resources and plays an active role in biodiversity conservation [35]. Topographic factors are important external environmental factors influencing changes in community species diversity [36]. In particular, plant diversity in the karst areas is influenced by factors such as altitude and slope [37]. In this study, the overall species diversity at medium altitude was higher than that at lower and higher altitudes (except for the diversity of the shrub layer at higher altitudes in the range of 1100–1160 m). This indicates that it increases with altitude and begins to decline after reaching a maximum at mid-altitude, showing a so-called single-peaked distribution pattern, which is formed by strong anthropogenic disturbance at lower altitudes [38]. Other studies along elevational gradients have found similar findings [39]. From the actual survey, the lower elevations of the mountain are highly influenced by human activities due to their proximity to settlements, and most of the native vegetation has been destroyed. This study shows that most of the spatial structure indices and species diversity vary significantly under different altitude gradients, which is mainly due to the fact that there are often heterogeneous changes in solar radiation intensity, soil nutrients, precipitation, and other hydrothermal environmental factors in different altitude gradients in karst areas. Additionally, changes in altitude gradients affect plant growth and development and cause complex adaptive changes in plant traits, resulting in plants forming different functional trait [40]. Furthermore, previous studies have shown that the strong relationships between climate, topographic factors, and plant richness, are mainly impacted by temperature variations as well as latitude and altitude effects [41,42]. The reason is that, in the lower and higher altitude villages, the influence of human activities is greater, and in recent years, the intensity of tourism development has led to poor soil and the single species composition of trees and shrubs; in the middle altitude, human interference is less, the combination of water and heat is the best, and the richness of trees and shrubs is the highest, which is consistent with the actual survey results. It can be seen that the altitude gradient influences a variety of ecological processes and the intensity of human activities, and the changes in species diversity patterns with different altitude gradients show the ecological characteristics of species, their adaptability to the environment, and the influence of human interference on biodiversity at different altitude gradients [43], which provides a reference basis and theoretical support for biodiversity conservation and sustainable development in mountain karst traditional villages. This study also shows that overall plant species diversity was highest in the living spaces, followed by the ecological spaces, and lowest in the production spaces. This is due to the fact that the formation of the community spatial structure is strongly related to the way of forest renewal and species competition [44], and has a certain correlation with traditional village culture and living customs, etc. In the living space, people keep the traditional customs, beliefs and so on, so that the plant landscape in the living space, such as the courtyard, is strictly protected by the villagers, and the degree of interference is the lowest, so that species diversity is improved. In the ecological space, in the process of growth and the succession of natural forests, culturally protected forests and so on, the same tree species has a self-thinning phenomenon due to fierce competition for limited environmental resources, which leads to the invasion of other species to form mixed tree species, and finally develops into a random distribution pattern [45]. While in the production space, the species of vegetable lands, tea gardens, farmlands, and fruit forests are relatively single, the species and quantity of plants are less, the community structure tends to be simple, and the species diversity is obviously reduced [46]. This further suggests that traditional village culture and its way of life have an important influence on village species diversity [47]. Therefore, village culture and its way of life should be taken into account when carrying out village planning. In summary, this provides a theoretical basis for the spatial planning of China’s land and the rational optimization of the layout of the “PLE spaces”.

### 4.2. Characteristics of the Relationship between Plant Community Structure and Species Diversity in Traditional Villages in Karst Topographic Regions of the Zunyi City and Development Strategies

At present, exploring the relationship between stand structure and understory vegetation species diversity has become a hot issue in forestry research [48]. Understanding the drivers of understory vegetation species diversity is of great significance to reveal the formation and maintenance mechanisms of understory vegetation species diversity [49]. The present study showed a coupled relationship between plant community spatial structure and plant community species diversity, which further quantitatively argues for the existence of a mapping relationship between stand spatial structure, the ecological environment, and species diversity. This study shows that the *W* of plant communities in traditional villages in karst topographic regions of Zunyi city as a whole has a highly significant negative correlation with the herb layer diversity indices *D* and *Jh*, and a significant positive correlation with the tree layer *Jh*, indicating that the more the spatial distribution pattern of the community tends to aggregate distribution, the lower the species diversity index of the herb layer, the more uniform the species distribution of the tree layer, that is, the larger the aggregation index, the more uniform the distribution of the forest. Correspondingly, the uniformity of understory shrubs will also be improved. This is close to the results of the study by Gengzhan Bian et al. [50] on the relationship between community structure and species diversity in secondary natural forests. At different altitudes, the *M* was significantly or very significantly positively correlated with species diversity in the tree layer (except at 700–800 m). In addition, the *M* was highly significantly correlated with shrub species diversity at the highest altitudes of 1100–1160 m and with herb species diversity *R*, indicating that the greater the *M* in the tree layer at higher altitudes, the higher the shrub and herb species diversity. This further indicates that the *W* in the tree layer is a key factor influencing species diversity, and the correlation between the *M* and the diversity of understory shrubs and herbs is greater [51]. This is basically consistent with the findings of Jun Zhu et al. [52] and Li Fang et al. [53]. At the lowest altitude of 600–700 m, there was a significant negative correlation between the *W* and the species diversity *D* of the herb layer, which indicated that the greater the individual distribution pattern of plants at low altitudes, the lower the diversity of the herb layer. This may be due to the grazing phenomenon in low altitude areas. The herb layer is relatively seriously disturbed by humans and livestock, and the increase in the number of tourists and other disturbances, resulting in the loss of some species in the herb layer and a decrease in species diversity [54]. At the highest altitude of 1000–1160 m, the *W* is significantly positively correlated with the species diversity *R* of the herb layer. Compared with low altitude, there was less human disturbance here, thus forming a peak area of species diversity, which means that high altitude is conducive to the growth of understory herbs. At lower altitudes of 700–800 m, all factors had a greater influence on plant species diversity (except for *W* and *M*), with a significant negative correlation between *U* and tree diversity *Jh*, suggesting that the greater the size differentiation at lower altitudes, the lower the tree diversity, probably due to severe anthropogenic disturbance at lower altitudes. According to the energy hypothesis, energy is an important factor affecting biodiversity. Species diversity changes are controlled by energy [55,56]. The *CI* was significantly negatively correlated with species diversity in the herb layer, probably due to the lack of nutrient space and living space in the stand as a result of intense competition between trees, which inevitably led to intense competition within and between herbaceous species in the stand, resulting in a reduction in the richness and diversity of herbaceous species in the stand [57]. Previous studies also found that competition between trees mainly occurred between neighboring trees, manifesting as competition between neighboring trees for light resources, soil resources and water resources within the competition unit. The exclusion effect of trees with a high level of competition exacerbates the disadvantageous position of understory herbs under limited light and water conditions, resulting in the elimination of some herbs by competition, thus reducing the species diversity of understory herbs [58]. The *DC* was highly significantly positively correlated with species diversity in the tree layer and negatively correlated with *Jh* in the herb layer, which indicated that the crown width and *DBH* of the tree species in the lower altitude area were large, and the residents had a strong awareness of their protection, so that they grew well. The growth status of the herb layer was closely related to the tree and shrub layer. The higher the *DC* of the tree, the lower the diversity of the herb layer. Furthermore, according to the survey, there was a custom of worshipping sacred trees in the village, such as the big banyan tree near the village, which was believed to bless peace, ward off disasters, and suppress diseases. Therefore, in the customary law, it is forbidden to cut down and the big banyan tree is strictly protected near the village. It was also a perfect presentation of traditional village folk culture. The *DH* is significantly negatively correlated with the herb layers *H* and *D*, which indicated that the community was more stable and more resistant to disturbance [59]. The tree community in this study is a largely developed community. The greater the tree diversity, the higher the stability of the community [60]; correspondingly, the greater the growth inhibition effect on the herb layer, results in lower species diversity in the herb layer [61]. In the *PLE* spaces, the *M* in the living and ecological spaces had the greatest impact on the species diversity of the herb layer, which is significantly or extremely significantly negatively correlated. This may be due to the greater the *M*, the more intense the competition among the tree species in the stand, the more fully the limited resources were utilized, and the growth of the shrub and herb layer under the forest was inhibited. It was consistent with the results of Guangyu Zhu et al.’s [62] study on oak natural secondary forests in Hunan. In the production space, the plant community structure was most closely related to species diversity, with all structural factors significantly or highly significantly correlated with tree and shrub species diversity (except for *U* and *CI*). Among them, the *W* was significantly or highly significantly positively correlated with the shrub layer species diversity, indicating that as the *W* of a stand increases, the degree of species diversity within the stand becomes higher, the community becomes more complex and the distribution of trees tends to be more heterogeneous [63]. The *DC* was significantly positively correlated with the tree layer species diversity and negatively correlated with the shrub layer. The *DH* was significantly positively correlated with the tree layer species diversity and significantly positively correlated with the herb layer species diversity. This indicates that there are many ornamental tree species in this space, and the villagers have a strong awareness of their protection. Such tree species have strong growth vitality, good stability, and good species diversity, thus inhibiting the growth of middle shrubs and reducing the species diversity of the shrub layer. In the living space, the *W* has the greatest impact on the species diversity of the herb layer, showing a very significant negative correlation, indicating that residents are more active in the living space, and may be reflected in the impact of artificial planting. Green space types with large human activities (such as farmland, vegetable land) have low species diversity in the herb layer. It can be seen that the plant species diversity of traditional villages in karst regions is mainly affected by natural conditions and human disturbance [64]. In the ecological space, the key factors affecting the species diversity of the herb layer were *W* and *DH*. The *W* was significantly or highly significantly negatively correlated with the species diversity of the herb layer, indicating that there are green space types with less human interference (forests, culturally protected forests, ancient trees, rivers, etc.), where trees grow better and are more stable, and species diversity is higher. The residents of traditional villages have formulated a strict management system for the forest, and no one is allowed to enter into the divine forest at will or cut down the divine trees of the forest [65]. Nowadays, the original forest around the village is in good condition, indicating that the binding force of customary law has played an important role. The size of the forest distribution pattern has a greater influence on the diversity of species in the herb layer.

In summary, there are many factors that influence the structure and diversity of traditional village plant communities, mainly the traditional culture of the village, its way of life, topography, altitude, and different spatial attributes. Therefore, to improve species diversity, we can adopt measures such as adjusting the structure of tree species and optimizing the spatial arrangement pattern of forest trees, and consider a comprehensive plan to adjust the spatial distribution pattern of forest trees. At the same time, from the characteristics of plant diversity in villages with different altitude gradients, it can be seen that the development of modern tourism has brought about adaptive changes to the ecological environment. From a systemic viewpoint [66], only by maintaining excellent stand spatial structure can the function of the village green space ecosystem be better played. The essence of the structural management of forest, farmland, and other green space, is to discuss the direct or indirect influence of stand spatial structure on stand function from the complex phenomenon put forward by the ideal stand spatial structure mode, and realize the multi-function management of the plant community [67]. Therefore, to improve community structure and diversity, tourism should be developed appropriately, human interference should be reduced, natural regeneration should be promoted, and integrated ecological management should be strengthened to ensure coordinated social, economic, and ecological benefits [68]. From the relationship between community structure and species diversity in the *PLE* spaces, it is clear that traditional villages are subject to the interplay of natural ecological protection, economic production development and social progress, and that the village ecological environment and its community spatial structure are highly susceptible to change due to human disturbance activities. So, the village ecological environment should be protected, and its individual distribution pattern characteristics can be fully considered according to the needs of different residential functions, such as the external environment of farm households (including courtyards, squares, and the side of the house) should be glorified and productive. The agricultural land (maize fields, rice fields, etc.) should be productive, and the ecological space (rivers, forests, etc.) should be adjusted in a targeted manner, with ecological conservation and biodiversity protection in mind. The above analysis shows that the karst topography and the village’s own long culture are inextricably linked to the plant landscape. This natural ecological view of the harmonious coexistence of people and nature reflects the ecological wisdom of the inhabitants of traditional villages in karst regions. The unique village topography and the ecological concept of village culture are the internal driving force of local residents’ environmental protection, which strengthens the protection of plant communities of traditional villages in karst regions, and reduces the intensity of human interference. Meanwhile, it is also the internal reason for the maintenance of the plant community structure and species diversity in traditional villages in karst regions, and also reflects that the protection and development policy of Chinese traditional villages is an important strategy with natural and cultural win-win value. Therefore, respecting the contractual spirit between humans and nature, focusing on the expression of humanistic and ecological values, and pursuing the harmonious coexistence between humans and nature for mutual benefit are important initiatives to improve community structure and species diversity. They are also worthy of reference for the development of the ecological environment in other rural villages in the context of rural revitalization.

## 5. Conclusions

This study confirmed the correlation between vegetation structure and species diversity. Vegetation structure had significant effects on plant diversity; the relationship between overall structural indicators of forest stands and the diversity indices is different. Specifically, W, M, U, and CI had significant effects on plant diversity, and the relationship between overall structural indicators of forest stands and the diversity indices of trees, shrubs and herbs are different. In addition, as for the different altitudes and “production-living-ecology” spaces, the structural indicators of forest stands were related to the diversity of trees, shrubs, and herbs. Furthermore, it proved that the plant community spatial structure is heavily influenced by natural conditions, such as topography and topography, and human disturbance activities in traditional villages in karst regions; the same two factors also have a great impact on plant community species diversity. The planning of the plant community structure and species diversity conservation should be coordinated, and the relationship between human activities and the natural environment should be regulated by taking appropriate measures to strengthen the internal links between different land types, and make them an organic whole. This study can provide a theoretical basis for future adjustments to the plant landscape of traditional villages in karst topographic regions of the Zunyi, on the basis of a comprehensive consideration of the spatial structure of plant communities and the conservation of plant diversity.

## Figures and Tables

**Figure 1 plants-11-03161-f001:**
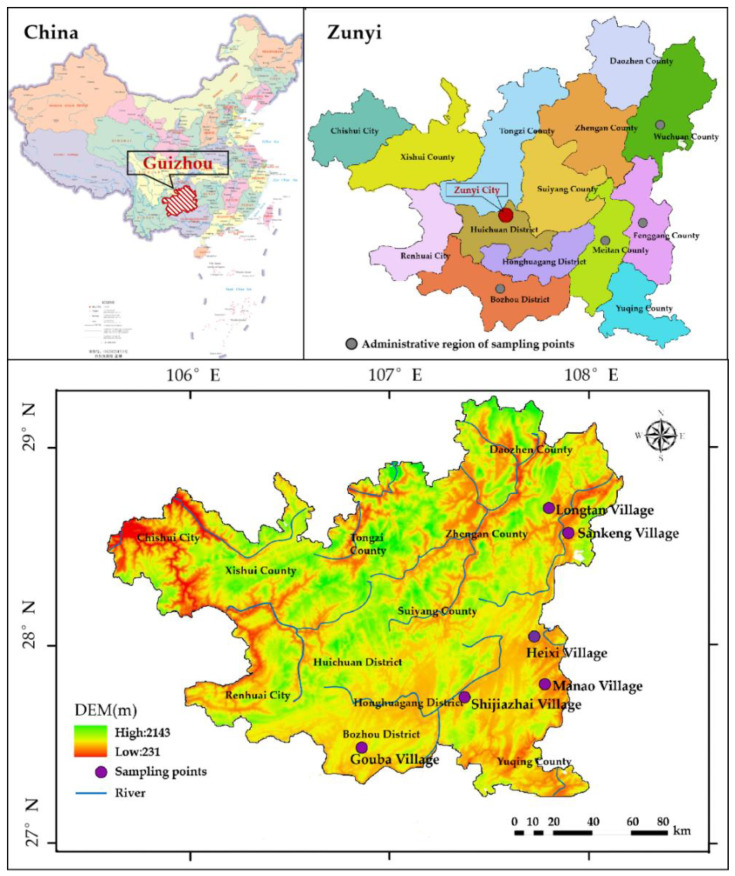
Study area and six traditional villages distribution.

**Figure 2 plants-11-03161-f002:**
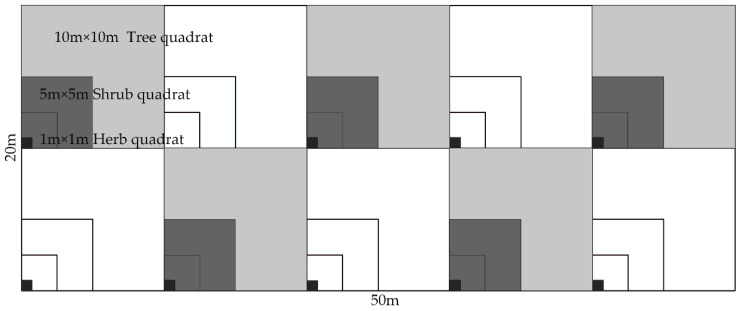
Plot setting.

**Figure 3 plants-11-03161-f003:**
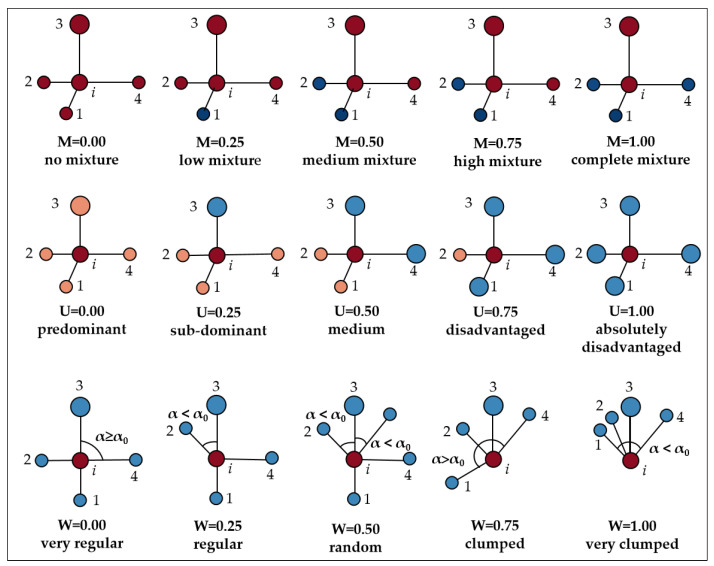
Indices of stand structure (*M*, *U*, and *W*) using nearest-neighbors spatial relationships based on a 5-tree group. Each tree *i* has a unique *M_i_ U_i_*, *W_i_* that takes one of the following values: 0.00, 0.25, 0.50, 0.75, 1.00.

**Figure 4 plants-11-03161-f004:**
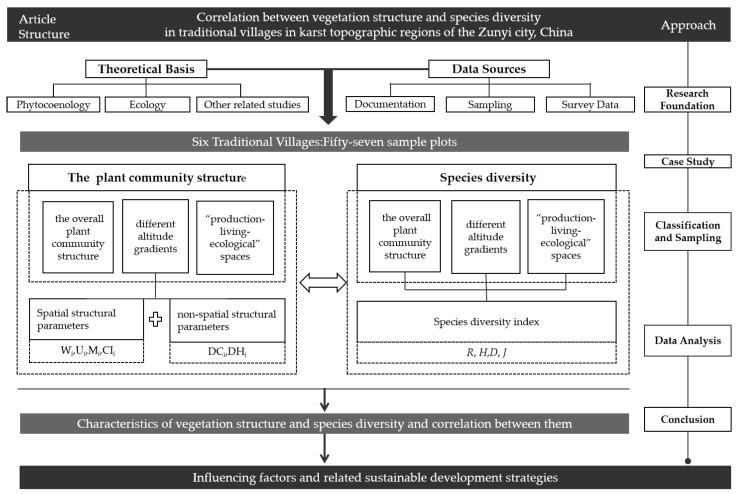
Research framework.

**Figure 5 plants-11-03161-f005:**
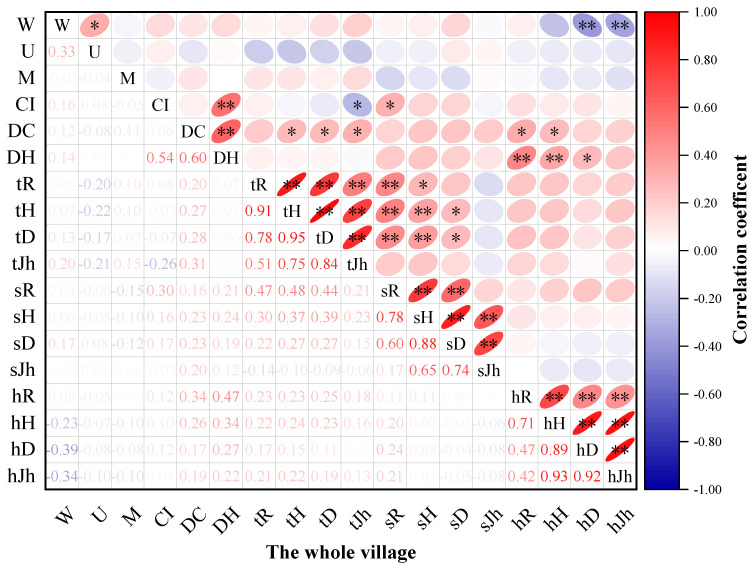
Correlation between plant community structure and species diversity in traditional villages in karst topographic regions of the Zunyi. Note: *R*: Margalef index; *H*: Shannon-Wiener diversity index; *D*: Simpson index; *Jh*: Pielou evenness index; *t*, *s* and h before diversity index are trees, shrubs and herbs respectively, e.g., *tR* is tree layer *R*, * indicates significant correlation (*p* < 0.05), ** indicates highly significant correlation (*p* < 0.01) (same below).

**Figure 6 plants-11-03161-f006:**
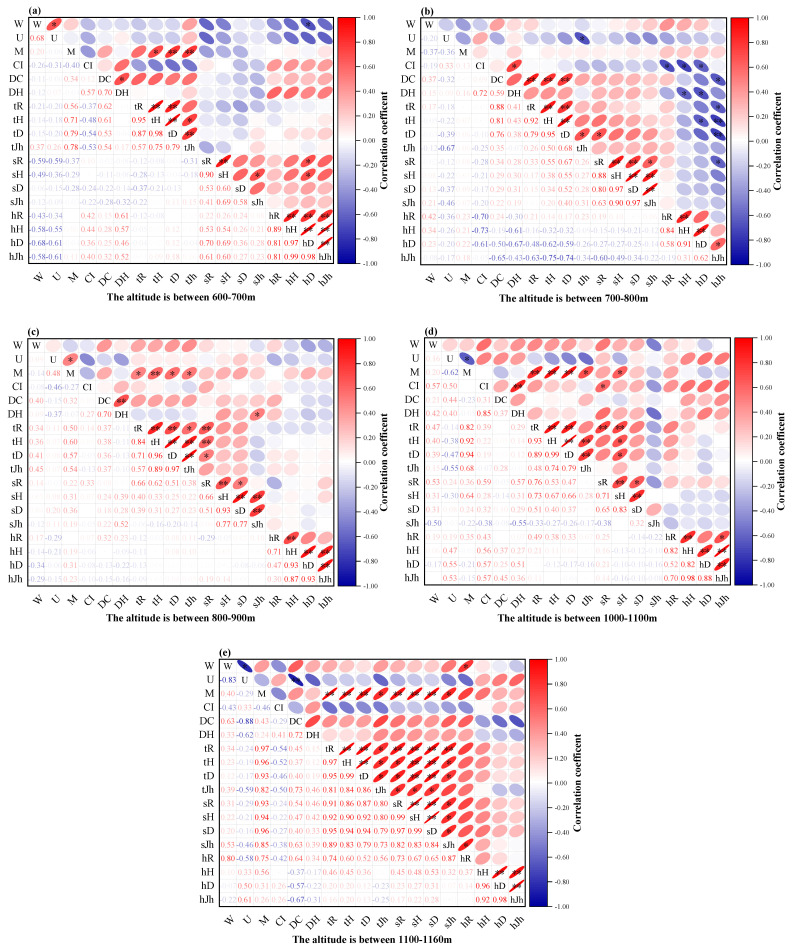
Relationship between plant community structure and species diversity in different altitude gradients. (**a**) The altitude is between 600–700 m; (**b**) the altitude is between 700–800 m; (**c**) the altitude is between 800–900 m; (**d**) the altitude is between 1000–1100 m; and (**e**) the altitude is between 1100–1160 m.

**Figure 7 plants-11-03161-f007:**
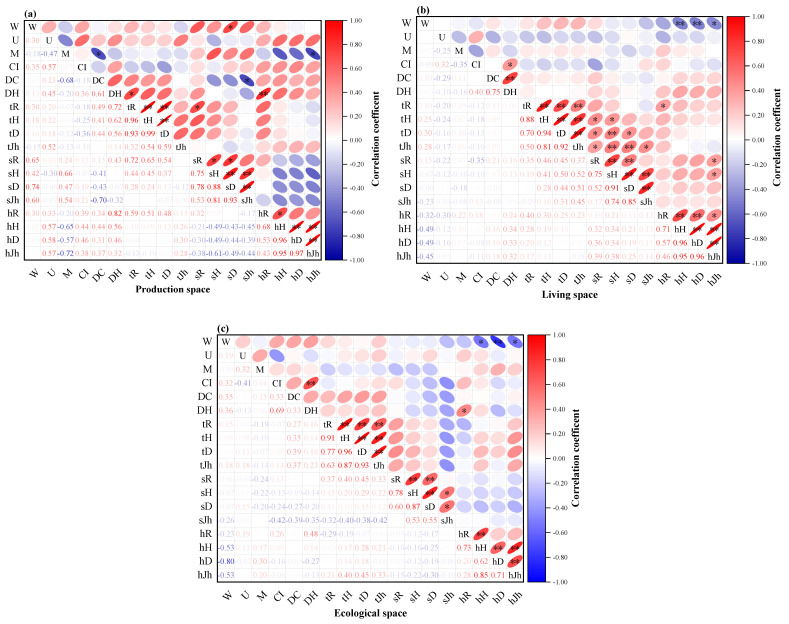
Relationship between plant community structure and species diversity in different altitude gradients. (**a**) Production space; (**b**) living space; and (**c**) ecological space.

**Table 1 plants-11-03161-t001:** Basic situation and representative characteristics of research villages.

Village Name	Longitude and Latitude	Nation	Area (hm^2^)	Population	Representative Features
Longtan	28°36′42″ N, 107°57′51″ E	Gelao, Miao nationality	31.71	436	It was one of the first Chinese traditional village. The natural pattern is “living based on the mountain, with the stone forest reflecting the mask”.
Sankeng	28°36′09″ N, 108°00′06″ E	Gelao, Miao nationality	0.67	560	It is the fifth batch of traditional Chinese villages. A long history and a profound cultural heritage. Most of the ancient wooden structure houses are well preserved.
Manao	28°06′51″ N, 107°44′40″ E	Han, Gelao nationality	19.99	148	In December 2012, Manao Village was included in the first batch of Chinese traditional village list.
Heixi	28°16′19″ N, 107°42′59″ E	Han, Gelao, Tujia nationality	0.40	4868	In November 2014, Heixi Ancient Village was selected as the third batch of traditional villages in China, and the ancient buildings are relatively well preserved.
Shijiazhai	28°06′36″ N, 107°25′42″ E	Han, Gelao nationality	3.99	3552	It is the second batch of traditional Chinese villages. Preserves the people’s commune buildings and large areas of traditional residential buildings.
Gouba	27°38′19″ N, 106°34′07″ E	Han, Gelao nationality	16.70	3412	It is the third batch of traditional Chinese villages. There are famous scenic spots such as Gouba Conference site, Gouba Conference Exhibition Hall, ingenuity garden, “Dream 1935” light and shadow show.

**Table 2 plants-11-03161-t002:** The number of targeted microhabitats and classifications under different altitude gradients.

Altitude (m)	Summation	Longtan	Sankeng	Manao	Heixi	Shijiazhai	Gouba
600–700	10	10	——	——	——	——	——
700–800	10	——	——	——	5	5	——
800–900	18	——	——	9	5	4	——
1000–1100	12	——	7	——	——	——	5
1100–1160	7	——	3	——	——	——	4

**Table 3 plants-11-03161-t003:** The number of targeted microhabitats and classifications under the PLE spaces.

PLE Spaces	The Type	The Number	Longtan	Sankeng	Manao	Heixi	Shijiazhai	Gouba
		57	10	9	9	11	9	9
Produce	Vegetable field	6	1	2	——	1	1	1
	Tea garden	2	——	——	1	——	1	——
	Farmland	2	——	1	——	——	1	——
	Fruit forest	2	1	——	——	——	——	1
Living	Courtyard	6	1	1	1	1	1	1
	House side	7	1	1	1	2	1	1
	Roads	7	1	1	2		1	2
	Square	2	1	——	——	1	——	——
	Sacrifice	5	1	1	1	1	1	——
Ecology	Forest	12	3	1	4	2	2	1
	River channel	6	1	1		2	1	1

**Table 4 plants-11-03161-t004:** Indicators of spatial and non-spatial structural parameters of forest stands [25,26].

Index	Formula	Value					Index Meaning
		0.00	0.25	0.50	0.75	1.00	
Uniform angle index (*W_i_*)	$W_{i}=\frac{1}{4}\overset{4}{\underset{i=1}{\sum }}Z_{ij}$	very regular	regular	random	clumped	very clumped	*W_i_*, *i* is the core wood, *j* is the adjacent wood. When the *j*th angle is smaller than the standard angle (α_0_ = 72°), *Z_ij_* = 1, but not *Z_ij_* = 0. *W_i_* = 0, very regular distribution. *W_i_* = 0.25, regular distribution. *W_i_* = 0.50, random distribution. *W_i_* = 0.75, clumped distribution. *W_i_* = 1.00, very clumped distribution.
Dominance (*U_i_*)	$U_{i}=\frac{1}{4}\overset{4}{\underset{i=1}{\sum }}K_{ij}$	predominant	sub-dominant	medium	disadvantaged	Absolutely disadvantaged	*U_i_*, when the core wood *i* is more adjacent than the *j* strain. *K_ij_* = 1, and vice versa, *K_ij_* = 0. *U_i_* = 0, in the advantage. *U_i_* = 0.25, which is the sub-advantage. *U_i_* = 0.50, in Medium. *U_i_* = 0.75, in the survival of the fittest. *U_i_* = 1.00, in the absolute survival of the fittest.
Mingling (*M_i_*)	$M_{i}=\frac{1}{4}\overset{4}{\underset{i=1}{\sum }}V_{ij}$	no mingling	weak mingling	moderate mingling	high mingling	very high mingling	*M_i_*, when core wood *i* is not adjacent to the *j* th strain. *V_ij_* = 1, and vice versa, *V_ij_* = 0. *M_i_* = 0, no mingling. *M_i_* = 0.25, weak degree mingling. *M_i_* = 0.50, moderate mingling. *M_i_* = 0.75, high mingling. *M_i_* = 1.00, very high mingling.
Competition Index (*CI_i_*)	$CI_{i}=\overset{n_{i}}{\underset{i=1}{\sum }}\frac{D_{ij}}{D_{ij}\;*\;DIS_{ij}}$	——	——	——	——	——	Competition index *CI_i_* describing the size of forest tree competitiveness. *D_ij_* represents the chest diameter of adjacent wood *j*, and *D_i_* indicates the chest diameter of core wood *i*, *DIS_ij_*, it ote the European distance between core wood *i* and adjacent wood *j*.
Growth Vitality (*DC_i_*)	$DC_{i}=\frac{CW_{i}}{D_{i}}$	——	——	——	——	——	The *DC_i_* that determines the growth and vitality of forest trees. *CW_i_* and *D_i_*, the crown length and chest diameter of the core wood *i*, respectively.
Stability (*DH_i_*)	$DH_{i}=\frac{H_{i}}{D_{i}}$	——	——	——	——	——	The *DH_i_* that reflects the tree stability, *H_i_* and *D_i_*, tree height and chest diameter of the core wood *i*, respectively.

Where *i* is the reference wood (*i* = 1, 2, 3, …, *n*), *n* is the number of individual reference wood in the sample, and *j* is the adjacent wood (*j* = 1, 2, 3, 4).

**Table 8 plants-11-03161-t008:** Species diversity index of overall plant community in traditional villages in karst topographic regions of the Zunyi City.

Distribution Level	The Margalef Index (*R*)	The Shannon-Wiener Index (*H*)	The Simpson Index (*D*)	The Pielou Index (*J*)
Tree layer	2.330 ± 1.101	0.746 ± 0.272	0.720 ± 0.195	0.345 ± 0.067
Shrub layer	2.510 ± 1.196	0.830 ± 0.252	0.753 ± 0.158	0.350 ± 0.071
Herb layer	4.151 ± 1.122	1.083 ± 0.285	0.820 ± 0.173	0.324 ± 0.077
average value	2.996 ± 1.401	0.886 ± 0.304	0.764 ± 0.180	0.340 ± 0.072

**Table 9 plants-11-03161-t009:** Species diversity index of plant communities at different altitude gradients.

Altitude Gradients (m)	The Margalef Index (*R*)			The Shannon-Wiener Index (*H*)			The Simpson Index (*D*)			The Pielou Index (*J*)		
	Tree Layer	Shrub Layer	Herb Layer	Tree Layer	Shrub Layer	Herb Layer	Tree Layer	Shrub Layer	Herb Layer	Tree Layer	Shrub Layer	Herb Layer
600–700	1.811 ± 0.840a	2.500 ± 1.334ab	4.000 ± 1.200a	0.643 ± 0.240a	0.832 ± 0.272a	0.930 ± 0.295a	0.671 ± 0.190ab	0.770 ± 0.113a	0.723 ± 0.180a	0.340 ± 0.055ab	0.360 ± 0.051a	0.280 ± 0.080a
700–800	2.910 ± 1.393b	2.610 ± 0.900ab	4.330 ± 1.444a	0.873 ± 0.264b	0.810 ± 0.263a	1.182 ± 0.273b	0.804 ± 0.120b	0.726 ± 0.220a	0.870 ± 0.161b	0.376 ± 0.033a	0.310 ± 0.091a	0.360 ± 0.067b
800–900	2.900 ± 2.840b	2.780 ± 1.323a	4.390 ± 1.076a	0.865 ± 0.200b	0.880 ± 0.260a	1.200 ± 1.180b	0.800 ± 0.140b	0.764 ± 0.164a	0.884 ± 0.076b	0.356 ± 0.054ab	0.350 ± 0.085a	0.355 ± 0.044b
1000–1100	1.640 ± 0.860a	1.900 ± 1.070b	3.880 ± 0.901a	0.560 ± 0.300a	0.730 ± 0.233a	1.030 ± 0.320ab	0.572 ± 0.270a	0.704 ± 0.151a	0.800 ± 0.193ab	0.315 ± 0.112b	0.350 ± 0.058a	0.309 ± 0.086ab
1100–1160	2.300 ± 0.683ab	2.900 ± 1.030ab	4.124 ± 0.900a	0.820 ± 0.190b	0.950 ± 0.181a	1.113 ± 0.170ab	0.800 ± 0.123b	0.840 ± 0.080a	0.855 ± 0.076ab	0.370 ± 0.035ab	0.374 ± 0.012a	0.330 ± 0.051ab

**Table 10 plants-11-03161-t010:** Species diversity index of plant community in “production-living-ecological” spaces.

“PLE Spaces”	The Margalef Index (*R*)			The Shannon-Wiener Index (*H*)			The Simpson Index (*D*)			The Pielou Index (*J*)		
	Tree Layer	Shrub Layer	Herb Layer	Tree Layer	Shrub Layer	Herb Layer	Tree Layer	Shrub Layer	Herb Layer	Tree Layer	Shrub Layer	Herb Layer
Produce	1.540 ± 0.740a	1.437 ± 0.656a	3.806 ± 1.330a	0.580 ± 0.246a	0.664 ± 0.232a	0.858 ± 0.370a	0.650 ± 0.198a	0.680 ± 0.204a	0.658 ± 0.253a	0.377 ± 0.068a	0.350 ± 0.088a	0.256 ± 0.103a
Living	2.730 ± 0.900b	2.803 ± 1.089b	4.430 ± 1.150a	0.840 ± 0.223b	0.882 ± 0.224b	0.171 ± 0.245b	0.780 ± 0.144b	0.791 ± 0.120b	0.862 ± 0.25b	0.360 ± 0.051a	0.346 ± 0.057a	0.345 ± 0.062b
Ecology	2.260 ± 1.315b	2.815 ± 1.257b	3.950 ± 0.811a	0.720 ± 0.307ab	0.866 ± 0.268b	1.105 ± 0.180b	0.676 ± 0.242ab	0.746 ± 0.167ab	0.870 ± 0.080b	0.313 ± 0.080b	0.340 ± 0.082a	0.340 ± 0.045b

## Data Availability

Not applicable.

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
