# Peer review of "Correlation between Vegetation Structure and Species Diversity in Traditional Villages in Karst Topographic Regions of the Zunyi City, China"

_plants, 2022, doi:10.3390/plants11223161_

Round 1

Reviewer 1 Report

Review Comments

Title: Correlation between plant community structure and species diversity in traditional karst villages in Zunyi city

By Deng et al., #plants-1984492

General comments

The authors targeted to investigate the correlation between vegetation structure and species diversity in the six villages of Zunyi city, China probably having some unique culture, traditions, and karst topography. This MS has plenty of issues especially linked to presentation of the contents, and need a thorough major revision prior to publication. After going through the entire length of the MS, I think, it is not possible and fair for a reviewer to highlight and mention each and every point that need revision. Therefore, I am going to mention the main concerns only, but the authors are requested to go through the entire MS same way.

Specific comments

1.      The title need revision as it has presentation issue e.g. what do you meant by “Karst Villages”?? Karst is a type of unique topography. You can probably write as “Correlation between vegetation structure and species diversity in a karst topographic landscape of the Zunyi city, China”

2.      Again, in the first sentence of the abstract, quoting, “In order to understand the relationship between the spatial structure of plant communities and species diversity in karst traditional villages, a study was conducted on the relationship between plant community structure and species diversity in six typical karst traditional villages in Zunyi.” Do not know, why the authors are repeating the same words in a single sentence again and again. This is a major issue with this MS and need a THOROUGH REVISIT. You can write AS “The study of relationship between vegetation structure and diversity is important in an area having karst topography and unique traditional customs. A total of six villages in the Zunyi city, China were selected to collect the vegetation data……”.

3.      After above, add a sentence in the abstract depicting the need of this study with novel objectives and utilized M&M.

4.      In the abstract, the authors quoted “than those in natural forests”. For such presentation, at first, you need to tell about the number of targeted microhabitats, and how did you classified them, if otherwise, the readers would have no clarity.

5.      In the abstract, “The living space communities??” Or “The recorded plant communities in the living space”, similarly, “the production space was randomly distributed???” Or “The plant communities were randomly distributed in the production space”.

6.      Same issue again “The overall plant community The overall spatial structure of plant communities”

7.      Almost every phrase has some issues, that’s why, with this kind of presentation, it is not possible for a reviewer to even judge the scientific soundness of the MS contents.

8.      The first sentence of the introduction section say “With the rapid advancement of industrialization and urbanization, the urbanization and homogenization” repetition and repetition and repetition of the same words…..look at this too….” to expand the direction of flora research, to deepen the theory of flora research,”

9.      The authors mentioned that there are about 3820 plant species flourishing in the study area (please add reference), however, did not convey what number and types (i.e. herbs, shrubs, tree) they recorded in this survey at different sampling sites/villages. You can add the same as a supplementary data.

10.  If all the study villages have karst topography then why the authors added a column with heading “village landform” in table 1.

11.  How the 57 plots were distributed among the so many mentioned microhabitats??? I mean, it’s not clear that how many plots were studied by the authors in each villages considering 3 major spaces plus 5 elevation gradient plus 11 types of green spaces etc. Can you please add a GIS map displaying all the study locations and their classification based on microhabitat differences?

12.  Include references of each index in Table 2.

13.  The authors should at least do rotation/switch axes of the tables for better readability, where possible. Hence, you would also be able then to even merge the two tables like table 4 and 7 as a single table. Same can be done with others.

14.  Why the authors did not perform hierarchical classification and ordination (say canonical correspondence analysis) of the studied vegetation samples and their determinants say elevation, aspect, slope etc.?

15.  Discussion and conclusions are superficial. The results are repeated in the conclusion section again and not representing the true study conclusion.

16.  Even all the quoted references in the list are not alike, and as desired by MDPI.

17.  This MS need a thorough revision, and services of a native English language expert.

Author Response

Reviewer (1)

Dear Reviewers:

We would like to thank the reviewers’ for giving us constructive suggestions which would help us both in English and in depth to improve the quality of the paper. Here we submit a new version of our manuscript with the title “Correlation between vegetation structure and species diversity in traditional villages in karst topographic regions of the Zunyi city, China” which has been modified according to the reviewers’ suggestions. Efforts were also made to correct the mistakes and improve the English of the manuscript. We mark all the changes in blue in the revised manuscript.

The following is a point-to-point response to the reviewers’ comments.

Comment 1:

The title need revision as it has presentation issue e.g. what do you meant by “Karst Villages”?? Karst is a type of unique topography. You can probably write as “Correlation between vegetation structure and species diversity in a karst topographic landscape of the Zunyi city, China”

Response 1:

Thank the reviewers for the comment. Your comment motivated us. Your suggestion that the tittle “Correlation between vegetation structure and species diversity in a karst topographic landscape of the Zunyi city, China” is very good. It should be noted that considering the particularity of our research object, our research object is mainly traditional villages. Traditional villages, also termed ancient villages, with both material and intangible cultural heritage. They have important historical, cultural, scientific, artistic, social, and economic value. Therefore, according to your valuable comments, we will write the title as “Correlation between vegetation structure and species diversity in traditional villages in karst topographic regions of the Zunyi city, China”.

Comment 2:

Again, in the first sentence of the abstract, quoting, “In order to understand the relationship between the spatial structure of plant communities and species diversity in karst traditional villages, a study was conducted on the relationship between plant community structure and species diversity in six typical karst traditional villages in Zunyi.” Do not know, why the authors are repeating the same words in a single sentence again and again. This is a major issue with this MS and need a THOROUGH REVISIT. You can write AS “The study of relationship between vegetation structure and diversity is important in an area having karst topography and unique traditional customs. A total of six villages in the Zunyi city, China were selected to collect the vegetation data……”.

Response 2:

Thank you for reminding us of the improper description of the abstract. Maybe the English expression is incorrect. We have the improper parts revised accordingly.

Comment 3:

After above, add a sentence in the abstract depicting the need of this study with novel objectives and utilized M&M.

Response 3:

Thank the reviewers for the comments. As seen the description of the abstract in the manuscript. We have added the need of this study with novel objectives and relevant research methods.

Comment 4:

In the abstract, the authors quoted “than those in natural forests”. For such presentation, at first, you need to tell about the number of targeted microhabitats, and how did you classified them, if otherwise, the readers would have no clarity.

Response 4:

Thank you for reminding us of vague descriptions of the number of targeted microhabitats and classifications. As seen the description of the section"2.2.2. Determination of parameter values for each analytical perspective", we described in detail the number of targeted microhabitats and classifications. And we have added 2 forms (Table 2 and Table 3).

Comment 5:

In the abstract, “The living space communities??” Or “The recorded plant communities in the living space”, similarly, “the production space was randomly distributed???” Or “The plant communities were randomly distributed in the production space”.

Response 5:

Thank you for reminding us of the improper description of the study. We have the improper parts revised accordingly.

Comment 6:

Same issue again “The overall plant community The overall spatial structure of plant communities”.

Response 6:

Thank you for reminding us of the improper description of the study. We have the improper parts revised accordingly.

Comment 7:

Almost every phrase has some issues, that’s why, with this kind of presentation, it is not possible for a reviewer to even judge the scientific soundness of the MS contents.

Response 7:

Thank you for reminding us of the improper description of the study. Maybe the English expression is incorrect. We have the improper parts revised accordingly. The English writing in this manuscript has been improved to eliminate the mistakes of grammar and syntax, and the description of the work has been refined to enhance the clarity. We hope that this new manuscript will be convincing.

Comment 8:

The first sentence of the introduction section say “With the rapid advancement of industrialization and urbanization, the urbanization and homogenization” repetition and repetition and repetition of the same words…..look at this too….” to expand the direction of flora research, to deepen the theory of flora research,”

Response 8:

Thank you for reminding us of the improper description of the study. We have the improper parts revised accordingly. On expanding the direction of flora research, to deepen the theory of flora research. We are very sorry. Considering the length of the description of the two subjects of plant community structure and diversity is too large, we decided not to describe the flora in detail, otherwise the introduction would be too complicated.

Comment 9:

The authors mentioned that there are about 3820 plant species flourishing in the study area (please add reference), however, did not convey what number and types (i.e. herbs, shrubs, tree) they recorded in this survey at different sampling sites/villages. You can add the same as a supplementary data.

Response 9:

Thank the reviewers for the comments. We added the source (http://www.lyj.zunyi.gov.cn/),and we submitted the same as a supplementary data file(See file compression package:Data)

Comment 10:

If all the study villages have karst topography then why the authors added a column with heading “village landform” in table 1.

Response 10:

Thank the reviewer for the comments. We modified this part by deleting.

Comment 11:

How the 57 plots were distributed among the so many mentioned microhabitats??? I mean, it’s not clear that how many plots were studied by the authors in each villages considering 3 major spaces plus 5 elevation gradients plus 11 types of green spaces etc. Can you please add a GIS map displaying all the study locations and their classification based on microhabitat differences?

Response 11:

Thank the reviewer for the comments. We modified this part by drawing a GIS diagram.

Comment 12:

Include references of each index in Table 2.

Response 12:

Thank the reviewer for the comments. We have added references to the indicators in table 2.

Comment 13:

The authors should at least do rotation/switch axes of the tables for better readability, where possible. Hence, you would also be able then to even merge the two tables like table 4 and 7 as a single table. Same can be done with others.

Response 13:

Thank the reviewer for the comments. We revised the tables of manuscript accordingly. The revised tables add the readability of manuscript. We did rotation for them.

Comment 14:

Why the authors did not perform hierarchical classification and ordination (say canonical correspondence analysis) of the studied vegetation samples and their determinants say elevation, aspect, slope etc.?

Response 14:

Thank the reviewer for the comments. We’re sorry for making you misunderstanding. Our view is as follows.

  1. About the canonical correspondence analysis. Considering that the analysis of the two main parts of the community structure and diversity in the manuscript is too complicated, we decided to omit the canonical correspondence analysis of this part.
  2. About their determinants say elevation, aspect, slope etc. We analyzed the effects of altitude on community structure and species diversity. Compared with altitude, the slope and slope of the survey plot did not change much, so we did not analyze the two factors of slope and slope.

Comment 15:

Discussion and conclusions are superficial. The results are repeated in the conclusion section again and not representing the true study conclusion.

Response 15:

Thank you for reminding us of the superficial description of the discussions and conclusions. We revised this part by reading the latest papers.

Comment 16:

Even all the quoted references in the list are not alike, and as desired by MDPI.

Response 16:

Thank the reviewer for the comments. We have corrected all references cited in the list and met MDPI requirements.

Comment 17:

This MS need a thorough revision, and services of a native English language expert.

Response 17:

Thank the reviewer for the comments. We made a thorough revision of MS. And We have sought the help of teachers whose major is English. If there are still serious deficiencies, we will amend it again.

Other corrections:

  • The Keywords “karst traditional villages” is replaced by “traditional villages in karst regions”
  • Section"2.3 Sample plot setting and survey", we added a plot setting diagram (Figure 2).
  • Re-adjust the picture clarity (Figure 5, Figure 6, Figure 7)

We have the improper parts revised accordingly. The English writing in this manuscript has been promoted to eliminate the mistakes of grammar and syntax, and the description of the work has been refined to enhance the clarity.

We tried our best to improve the manuscript and made some changes in the manuscript. The quality and clarity of the manuscript have been improved a lot according to your specific and valuable comments. We appreciate for Reviewers' warm work earnestly and hope that the correction will meet with approval. If there are still questions in the correction, we would be delighted to receive your comments.

Once again, thank you very much for your comments and suggestions.

Best wishes,

Zongsheng Huang

Reviewer 2 Report

Manuscript "Correlation between plant community structure and species diversity in traditional karst villages in Zunyi city" is very interesting.

Authors provided a theoretical basis for the maintenance of traditional village
botanical landscapes in Zunyi and the establishment of rural botanical landscapes in
other karst areas, and thus provide theoretical guidance for the sustainable development
of village biodiversity under the strategy of rural revitalisation and urbanisation in
China.

Introduction and Discussion are perfect.

Section "2.3. Data processing" is poor.
Lack information about distribution.

Quality of Tables is very poor. Tables need correction.

Authors should use multivariate methods.

Paper needs minor revision.

Author Response

Reviewer (2)

Dear Reviewers:

We would like to thank the reviewers’ for giving us constructive suggestions which would help us both in English and in depth to improve the quality of the paper. Here we submit a new version of our manuscript with the title “Correlation between vegetation structure and species diversity in traditional villages in karst topographic regions of the Zunyi city, China” which has been modified according to the reviewers’ suggestions. Efforts were also made to correct the mistakes and improve the English of the manuscript. We mark all the changes in blue in the revised manuscript.

The following is a point-to-point response to the reviewers’ comments.

Comment 1:

Introduction and Discussion are perfect.

Response 1:

Thank the reviewers for the comment.

Comment 2:

Section "2.3. Data processing" is poor. Lack information about distribution.

Response 2:

Thank the reviewer for the comments. We modified this part by drawing a Research Framework Diagram (Figure 4)

Comment 3:

Quality of Tables is very poor. Tables need correction.

Response 3:

Thank the reviewer for the comments. We revised the tables of manuscript accordingly. The revised tables add the readability of manuscript. We did rotation for them.

Comment 4:

Authors should use multivariate methods.

Response 4:

Thank the reviewer for the comments. We revised the of manuscript accordingly by use multivariate methods.

Comment 5:

Paper needs minor revision.

Response 5:

Thank the reviewer for the comments. We revised the tables of manuscript accordingly.

Other corrections:

  • The Keywords “karst traditional villages” is replaced by “traditional villages in karst regions”
  • Section"2.3 Sample plot setting and survey", we added a plot setting diagram (Figure 2).
  • Re-adjust the picture clarity (Figure 5, Figure 6, Figure 7)

We have the improper parts revised accordingly. The English writing in this manuscript has been promoted to eliminate the mistakes of grammar and syntax, and the description of the work has been refined to enhance the clarity.

We tried our best to improve the manuscript and made some changes in the manuscript. The quality and clarity of the manuscript have been improved a lot according to your specific and valuable comments. We appreciate for Reviewers' warm work earnestly and hope that the correction will meet with approval. If there are still questions in the correction, we would be delighted to receive your comments.

Once again, thank you very much for your comments and suggestions.

Best wishes,

Zongsheng Huang

Round 2

Reviewer 1 Report

Dear Authors

I appreciate your significant revision to improve the the MS quality and presentation. 

Regards